# Effectiveness of a Motor Intervention Program on Motivation and Learning of English Vocabulary in Preschoolers: A Pilot Study

**DOI:** 10.3390/bs9080084

**Published:** 2019-08-05

**Authors:** Rosario Padial-Ruz, Raquel García-Molina, Esther Puga-González

**Affiliations:** 1Department of Didactics of Musical, Artistic and Corporal Expression, University of Granada, 18071 Granada, Spain; 2Unit of Scientific Culture of the University of Granada, University of Granada, 18071 Granada, Spain; 3University of Granada, 18071 Granada, Spain

**Keywords:** second language learning, motor activity, gesturing, preschool children, motivation

## Abstract

(1) Background: Linking physical activity to the teaching of curricular contents provides numerous motivational and emotional benefits which improve academic performance and lead to the improvement and creation of healthy habits from an early age. (2) Method: The objective of the study is to analyze the effectiveness of a 5-week intervention program based on the use of a combined methodology of physical activity and gestures on motivation and vocabulary learning in English. The sample of children was aged from 4 to 7 years and was recruited from three children’s centers in Tegucigalpa, Honduras. A quasi-experimentalstudy was carried out using a pretest–posttest design in a sample (n = 88). (3) Results: Statistically significant results were obtained in the learning of words through the combined methodology of gestures and motor activity, compared to the traditional methodology used in the control group. (4) Conclusions: The main conclusions are that motor and expressive activities at an early age can be an effective motivational resource that promotes an increase in children’s physical activity time in the classroom. Further, it improves academic performance, producing a more effective learning of the vocabulary of a second language.

## 1. Introduction

The decrease in the physical activity (PA) levels of children from an early age [1] is a relevant concern worldwide. Many children currently do not comply with the minimum requirement of 60 min of moderate or vigorous physical activity a day that is required for health [2]. According to UNESCO [3], neither do they achieve at least 180 min of weekly physical activity. This decrease is associated with a low mastery of motor skills within preschoolers [1,4] and an increase in sedentary lifestyle which causes a consequent increase in physical diseases, such as that seen in the alarming increase of childhood obesity [4,5]. Further, other mental or affective-emotional diseases become more likely [6,7,8], which means a decrease in the quality of life of these children in the future [9,10].

In addition to health benefits, there is a positive relationship between physical activity and academic performance. This makes it an effective tool in teaching for the development of cognitive content [11,12,13] since play and movement are significant resources for students. In this way attention and motivation towards learning is improved [14].

Engagement in physical activity produces an improvement in behavior [15,16] and in cognitive functions such as attention [17]. As a consequence, academic performance can improve [12,14,18,19].

Since children spend most of their time at school, this context presents a great opportunity for improvement and to increasephysical activity [15]. Opportunities for physical activity at school ariseduring the physical education session [20,21], through the teaching of other curricular contents [15,22], and during the breaks that occur during the school day [9,23,24,25].

However, several studies show that there is a lack of motor skill sessions taking place at school, whilst there is also a lack of preschool physical activity programs for young people [26,27,28,29]. Motor education is not treated with the importance it requires at these ages [30,31,32], since the majority of school time is dedicated to academic subjects which form the classic curriculum. This means that an opportunity is lost to capitalize on an environment that is conducive to motor development, better academic performance, the creation of healthy habits, and the prevention of a sedentary lifestyle and the diseases associated with it.This situation is aggravated in contexts characterized by a low socioeconomic status due to school absenteeism in this type of context [33]. This is the type of context that will be examined in this study.

Scientific evidence of the benefits that delivering active interventions based on a transversal axis of play and movement have on students (motivation, interest, autonomy, development of social skills, emotional and creative development, development of the physical condition, motor skills, and prevention of sedentary lifestyle) [34], show the need for planning and designing programs and activities based on physical activity and movement. Selected activities should be perceived by children as meaningful and enjoyable, enabling them to fully develop their skills in a motivating and physically active environment [35].

For learning to be effective teaching methods in these ages, it must be based on active, playful, action-oriented, and multi-sensed learning [36,37,38,39]. This is so they could simultaneously stimulate various sensory functions such as visual, auditory, tactile, and kinesthetic, thus integrating all psychomotor functions [37,40].

Conceptual development is closely linked to motor development and experience, so the basis of learning and the creation of meaning is based on sensory and motor participation [39]. Learning activities that contain movement alongside another activity, such as active stories and active songs, activities that are accompanied by significant gestures and rhythmic activities [41,42], are associated with emotions that facilitate more meaningful learning [31,43,44]. Through the involvement of several senses, a greater number of associations to the element being learned are produced, improving learning and retention [37].

In this line, in recent years, several studies have been carried out which sought to integrate physical activity into other different curricular contents such as Mathematics, Language, Science, or English [45,46,47,48]. These have highlighted the improvement that occurs to the academic performance of students, amongst other benefits. According to Bolaños [49] and Alcedo and Chacón [50], the linkage of education through physical activity with other disciplines of the curriculum leads this fusion of content to achieve academic improvements, increasing the quality and acquisition of learning in students.

Although there is limited research in the field of physical education, enhanced communication using multisensory teaching has been shown to provide opportunities for learning in students. Specifically, in the case of learning a second language at an early age, contemporary studies have implemented different methods, such as those based on multisensory learning [51]. For example, five methods for learning have been identified:(a) “The Good Start Method for English” method is aimed at stimulating the different senses (visual, auditory, motor, and tactile) in 5- to 7-year-old children. This is done by combining language learning with visual elements, graphics, and motor activity. For example, some songs refer directly to physical activities within the content of the song. Children can perform these activities while they sing. This improves memorization of verbal content and increases the willingness to learn [37,38]. (b) The “Total Physical Response Technique” method [52], which basically consists of listening to slogans and representing them physically, as well as using gesturesand simulated actions. More recently, other authors such as Al Harrasi [53] also introduced active songs. Results in children and adults (individuals aged 8 years and older), confirm a greater retention and comprehension of vocabulary following engagement with the singing game and thephysical response it entails.

(c) Methods based on the development of rhythmic abilities [42,54]. Activities and motor tasks such as songs, poems, and rhymes, favor the development of rhythmic abilities and skills that are related to phonological awareness and literacy skills [55]. These function boththrough the motivation that they promote within the child and via the auditory rhythmic processing they stimulate, which translate to the learning of other languages enabling them to be learned from an early age [42].

(d) Methods based on the use of gestures, pantomime, or facial expression [56,57]. Learning is based on the accompaniment of significant gestures which relate to the content being learned. An experiment using these methods was conducted by Albadalejo et al. [37] with 2 and 3-year-old children. They used songs and stories with accompanying gestures.The use of stories had for a more positive impact on vocabulary learning than the use of songs or a combination of both. In contrast, results obtained by Chlapana and Tafa [58] in a sample of immigrant children aged 4 to 6 yearsdid not demonstrate better vocabulary learning following involvement in a condition that used facial expressions and gestures. (e) The “Energizers” method [59] is designed to work on any curricular contents through physical activity. It consists of carrying out brief physical education sessions (10 min approx.) in the classroom. The sessions are directed by the teacher and they integrate academic content. Representation games and songs are also included.Studies conducted to examine this methodhave confirmed thatphysical activity programs introduced within the classroomare effective at increasing daily physical activity at school and improving taskbehavior during academic instruction. These studies have been found in [15]. Combining physical activities with task-relevant gestures also leads to better learning within childrenas young as 4 years old [22]. Physical activity programs delivered in the classroom such as the Energizersmethodare beneficial for children’s learning from an early age. This is the case both in terms of motivation towards a task and memory function, as well as increasingphysical activity in the classroom.Further, the scientific evidence confirms that the use of methodologies that combine motor activity with cognitive content learning have positive effects on learning (i.e., improved attention processes, better retention of content, and increased motivation towards learning through the use of playful and meaningful activities for students). The main objective of this study was to analyze the effectiveness of a 5-week intervention programwhich used a methodology combining physical activity and gestures, on the motivation and learning of English vocabulary. The sample used included children aged from 5 to 7 years old, from three children’s centers in Tegucigalpa, Honduras.

## 2. Materials and Methods

### 2.1. Subjects and Design

The sample of the present study consisted of 88 students, malesandfemales, with a minimum age of 4 and a maximum age of 7 years. Individuals came from three public schoolsin Tegucigalpa, Honduras, created by the NGO ACOES, where there is high school absenteeism and a low socioeconomic status. The sample was selected using convenience sampling and represents 100% of the population. Initially there were 94 students but 6 were excluded due to lack of attendance to intervention sessions, as a result of school absenteeism at these centers. The study carried out was adapted from and followed the basic guidelines of the intervention program carried out by Toumpaniari et al. [22]. A quasi-experimental designstudy was carried out using a pretest–posttest with a non-equivalent control group. The design is multifactorial and multivariate. The sample was divided into three non-equivalentgroups, due to the difficulty encountered in accessing the sample:Control Group (CG) (with 28 participants), Experimental Group 1 (EG1) (with 22 participants), and Experimental Group 2 (EG2) (with 38 participants).

The following inclusion criteria was considered:-Participants were required tobe aged between 4 and 7 years old.-Participants had to be attending one of the three public schools set up by the NGO.-Participants could not be suffering from any disease which could hinder the normal progress of the program.-Participants could not have any prior knowledge of asecond language, specifically English.

### 2.2. Measures

The following instruments were used for the collection of study data:Ad hoc questionnaire: For the registration of sociodemographic variables (gender, age, family situation, etc.).The smiley scale: To obtain the degree of satisfaction of the children with the methods used in the intervention. The Likert scale proposed by Jäger and Bortz [60] was used, modified, and applied to children. It presented two questions with three levels (“Yes”, “I do not know”, and “No”), which had to be responded to using emoticons:-Did you like the course?-Would you like to learn in this way in the future?Vocabulary Verification Checklist: Children were individually assessed to determine how many words from the previously studied vocabulary of 22 words relating to parts of the body they were able to remember. During completion of this test, the researcher named the body part in Spanish and the child said the corresponding word in English. The words were verified with a checklist that contained all the taught words. The researcher would put a cross in the “yes” box if a word was remembered correctly, or in the “no” box if the word was not remembered correctly [22].

### 2.3. Procedure

This research study followed the ethical principles for research established by the Declaration of Helsinki. Confidentiality of participants was respected at all times throughout the study and the ethical principles for medical research involving human subjects laid out in the Declaration of Helsinki (modification of 2008) were followed. Approval of these arch by the Human Research Ethics Committee of the University of Granada was requested, being sanctioned with code 462/CEIH/2017. Approval of school management was requested, and informed consent of parents or legal guardians was obtained for all participants.

The program was delivered between the months of August and September 2018. Intervention implementation in each of the two conditions consisted of two weekly sessions which lasted for 1 h during a period of 5 weeks and with a total of 10 sessions for each group.

Prior to commencing, each class was randomly assigned to one of the three study groups. This resulted in the establishment of a Control Group with n = 28 (traditional method), an Experimental Group 1 with n = 22 (gestures), and an Experimental Group 2 with n = 38 (gestures combined with motor activity) (Figure 1).

In order to preserve the anonymity of participating children and data confidentiality, an encoding of the sample was carried out. To do this, three categories corresponding to each treatment were established: Control Group (GC); Experimental Group Gestures (EGG); and Experimental Group Physical Activity and Gestures (EGPAG). Within each category a number was assigned to each individual participant (GC1–GC28, EGG1–EGG22, and EGPAG1–EGPAG38).

The sessions were given by one of the researchers and took place within the classrooms of each of the groups. The role of the teachers was merely supportive.

The content to be taught was composed of 22 words—substantives—in a foreign language (English), all belonging to the “Body Scheme” content block. The complete list of 22 words was worked on during each session.

The intervention process was as follows (Figure 2): Pretest: This was held duringthe first week of August. In all three conditions, the researcher began by naming a part of the body, verifying that none of the children in the group knew the translation into English before then proceeding to teach it by means of a flashcard. Treatment: After verifying that the three groups started at level 0, that is, they had no knowledge of the language, the specific activities corresponding to each condition were applied in each class.

Control Group: Flashcards were used to teach the words covered in class. The approach taken followed the online model (https://supersimple.com/free-printables/head-shoulders-knees-toes-flashcards/), which begins by presenting the image of a part of the body at the same time as introducing the spoken word in Spanish and then in English. For example, “Here we have the word” Cabeza–Head. “How do you say ‘head’ in English? Head.” In the control condition there was no specific activity, simply verbal repetition or representation through drawing.

Experimental Group 1: In the condition of teaching vocabulary through gestures, children repeated the name in both languages, as in the previous condition. They then made a gesture with that part of the body or simply pointed it out. At no time were motor activities used. For example, with the word “head”, the researcher named both words“Cabeza-Head” while showing the respective flashcard. The children were then instructed to nod their head but without moving from the place in which they were standing. The activity “It’s a zoo in here” was taken from the Energizers program [15] and songs including “Head, shoulders, knees andtoes”, and “Hokey-Pokey Shake” by Super Simple Songs were taken from the Internet (https://www.bing.com/videos/search? q = Super + Simple + Songs &qpvt = Super + Simple +Songs & FORM = VDRE). The songs were sung and gestures were made with the body parts mentioned by the songs. As with the previous activity, the children did not move from their place.

Experimental Group 2: In the condition of teaching vocabulary through gestures combined with motor activity, the activities of the “K2-Energizers” program were incorporated [15,61], [retrieved from https://gethealthyclarkcounty.org/wp-content/uploads/2017/08/cc-energizers.pdf]. An example of one of these activities is“Air writing”. This urges the students to move around the classroom as directed by their teacher. The teacher guides them to jump, walk, dance, turn, and move the body segment that is indicated. Specifically, in the condition of physical activity and gestures, children perform games and motor activities at the same time as gesturing with or to different parts of the body, as appropriate.

Posttest:Conducted at the end of the fifth week midway through September. An individual posttest was carried out with all the children in each group. For this, the smiley scale was used to measure the degree of motivation towards the different study methods used. The vocabulary checklist was also administered to determine how many of the 22 words were remembered by participants in each group.

### 2.4. Data Analysis

Statistical analysis was carried out using IBM SPSS^®^ 22.0 software (IBM Corp, Armonk, NY, USA). Frequencies and medians were used for the basic descriptive analysis. Relationships between variables were examined using contingency tables or cross tables.

## 3. Results

The descriptive results presented in Table 1 show that the average age of the sample was 5.45 years (SD = 0.693), with 45.5% (N = 40) being males and 54.5% (N = 48) being female. With regards to the family context of the sample, 47% of children indicated that they lived with both parents (nuclear family), whilst 38.6% were living in a single parent family situation (34.1% of the sample were living alone with the mother compared to 4.5% who lived only with the father). Finally, 13.6% were living with grandparents (extended family).

In consideration of the place where children played after school, it was noted that a low percentage of participants (34.1%) did so on the street, while 65.9% did so indoors. None of the participants (N = 88) engaged in extracurricular activities (sports, artistic, linguistic, etc.) (Table 1).

Table 2 reports the data obtained in the pretest. This demonstrates that 100% of the sample had no knowledge of the vocabulary targeted by the intervention prior to their involvement in the study. 

The relational analysis of the estimated variables was conducted by means of contingency tables. The following results were found:

The number of words recalled by participants was divided between three categories (0–6 words, 7–13 words, and 14–22 words). Examination of these three categories according to the three intervention conditions identified statistically significant differences (*p* < 0.001). Further perusal of the results shows that this difference is driven by the higher number of words learned by students in experimental conditions 1 and 2. Concretely, 68.4% of children in Experimental Condition 2 (Gestures + Motor Activity) learned 59.1% of the vocabulary taught and pertained to the maximum vocabulary group (14–22 words), whilst 54.5% of those who belonged to Experimental Condition 1 (Gestures), learned 27.3%. Those who participated in the control condition only learned 13.6% of the vocabulary (Table 3).

Regarding the degree of satisfaction with the approach taken in each condition, as can be seen in Table 4, no statistically significant differences were found. In fact, more than 90% of individuals in all three groups responded “Yes, I liked it” to both questions (1. Did you like the course? 2. Would you like to learn in this way in the future?).

## 4. Discussion

The main objective of this study was to analyze the effectiveness of a 5-week intervention programthatused a methodology of combining physical activity and gestures to motivate the learning of English vocabulary. The sample used were children aged from 5 to 7 years old, from three children’s centers in Tegucigalpa, Honduras. A comparison model was used comparing a traditional interventionand another based exclusively on the use of symbolic gestures.

As discussed throughout the introduction, several studies confirm the benefits of physical activity and the positive relationship between physical activity and academic performance [12,14,18,19]. This makes it an effective tool in teaching for the development of cognitive content [11,12,13,15,22,45,46,47,48,49,50] since play and movement are significant resources for children. In this way, attention [14,17] and motivation towards learning is improved [14,15,16]. Specifically, the use of programs based on games and movement within teaching of a second foreign language can improve the learning of cognitive contents, behavior, and health at early ages [15,22,34,38,41,42,46,58].

The development and application of neuroscience within the educational field provides conclusive data on the factors involved in the acquisition of a second language. One of the factors with most evidence is the age at which learning begins. Ideally, teaching should start at an early age (3–6 years or even earlier) as this is associated with important benefits. Young people demonstrate greater cerebral plasticity and so will have an accelerated cognitive development in comparison with those who start learning at later life stages [62]. Initiating learning earlier will therefore provide better results in general academic performance [14,57]. In addition, on a personal level, there is evidence of better communication, tolerance of other cultures, and higher self-esteem in general. In the specific learning of the language, young people have a greater capacity for listening and pronunciation, meaning they can bemore accurate and have better prospects for greater learning in the future [63]. Finally, in view of future employment and socioeconomic improvement possibilities, as our knowledge increases, the possibilities of professional opportunities and our potentialto obtain a decent job are broadened. These findings, along with our interest in the infant stage, motivated the decision to include the sample selected in the present study (88 children between the ages of 4 and 7).

Other important factorsthat influence the process of schooling and learning of children from an early age, and that relate to the context of the colleges participating in the present study, are the socioeconomic and cultural characteristics. This study was carried out within a disadvantaged context, characterized by low socioeconomic status, high rates of illiteracy, and school absenteeism. The centers involved in the present research were taken from a setting characterized by socioeconomical poverty. In such settings, the educational level of children is typically far below that found in other centers located within higher socioeconomic contexts. This is evidenced by the finding that none of the children had any prior knowledge of asecond language. Studies such as those conducted by Maguire et al. [64] support this assertion and highlight evidence of a low level of learning of vocabulary amongst children from disadvantaged contexts as opposed to others from more privileged economic contexts.

With regards to socioeconomic and cultural characteristics, family cohesion and family support can favor children’s learning, particularly in contexts of poverty [61,65,66]. The breakdown of familial relationships, such as when the sentimental relationship of parents ends, can convert this population into a vulnerable group [67,68]. Such characteristics were present in the study sample as indicated by the descriptive data analyzed. This is likely to inhibit their ability to learn.

The present research study also provides some context around the possibilities of play and development of motor skills. Students are often deprived of the possibility of playing or engaging in sport, both inside and outside of school. This is because they often lack time as they have to contribute to domestic chores, care for siblings, or help to obtain the necessary financial resources to survive by seeking employment [69]. Possibilities for sport or play also tend to be more limited at early ages [70]. This was shown by the descriptive analysis carried out in the present investigation.

In addition, neighborhoods where individuals of a low socioeconomic status tend to reside pose a high amount of danger due to the high rates of drug trafficking and violence. Thiscauses insecurity, which translates into decreased opportunities for play outside of school [69,70]. Deprivation of physical activity and motor development causes children from disadvantaged backgrounds to have lower levels of motor competence. In many cases they may fail completely to develop anything more than the most basic of fundamental movement skills [71]. Thus, in order to ensure that similar students in disadvantaged contexts are provided withthe opportunities to engage in play and to engage in motor activities, schools must seek to embed them within the design and planning of activities which combine play and motor activity with cognitive development and curricular contents overall [69]. This will enable an increase in the possibilities for integral development of students at early ages, since there will be an improvement in social, emotional, physical, and cognitive wellbeing.

As is seen in the present study, the integration of physical resources (gestures/rhythmic abilities) and motor activities within curricular contents for the teaching of English produces positive results with regards to learning and motivation of students at an early age. On the one hand, it facilitates the memorization and long-term maintenance of verbal content, and on the other hand, it contributes to basic motor development during youth.In the present study the use of gestures and, to an even greater degree, the combined use of gestures and motor activities such as games and active songs, obtained better results with regards to the motivation ofpre-school-aged students and their learning of vocabulary pertaining to a second language. This was in agreement with the guidelines presented by Toumpaniari et al. [22] in their similar study. Further studies that have rolled out similar interventions have also found results which corroborate those of the present research, in that motivation and performance of a memory test forvocabulary pertaining to a second language was improved relative to traditional teaching approaches. These other comparable interventions included rhythmic skills and gestures [37,38], rhymes and songs accompanied by gesturessimulation [42], activities and gestures [58], active games [46], and sports or physical bodily expression [72]. In contrast, the study carried out by Albadalejo et al. [41] demonstrated better vocabulary learning in 2- to 3-year-old students using stories and short stories rather than rhythmic methods in which songs were used either alone or in combination (stories and songs).

The analysis of data corresponding to the motivation of children receiving thedifferent intervention approaches delivered in the present study reflects a balance in the results, in that statistically significant outcomes were not found for the satisfaction questionnaire. This may be due to the fact that, as Milteer and Ginsburg [69] indicates, the drastic change in the type of activity carried out by the children and them being in an environment where they felt safer, provided a source of motivation in itself. Thus, regardless of the specific approach used, it seems that any type of intervention may enable disadvantaged children to escape from their personal and environmental situation. This coincides with the characteristics of the sample used in the present study, since the classes are made up of children who normally have a high level of school absenteeism. The NGO is responsible for removing them from their contexts and giving them an opportunity to receive a good basic education.

Some of the limitations of this study are that thesample wasselected according to convenience. More specifically, it was not possible to use students as the unit of randomization and so classes were randomized as a wholeto the experimental conditions. This meant that the groups did not contain equal numbers of students.Another limitation was the fact that a measure of fitness was not used. This makes it impossible to identify animprovement inchildren’s engagementin physical activity.

In addition, the two intervention conditions were implemented by a researcher from the study team carrying out the present research. It would be useful for future studies to train kindergarten teachers to implement these practices in their classrooms.

These considerations should be contemplated in future research.

## 5. Conclusions

The main conclusions of the study are discussed below. 

The review carried out prior to the design of the intervention indicated that the individual’s age at the initiation of learning was a key factor to impact upon the acquisition of a second language. Commencing learning activities prior to three years of age was found to be important due to greater cerebral plasticity. This results in an increase in cognitive development and, consequently, in general academic performance.

Socioeconomic and cultural characteristics are another important factor. Contexts of poverty, with unstructured family environments, promote a higher degree of school absenteeism at early ages an onwards, and consequently, lower academic performance. With regards to motor or physical aspects, the lack of access to play and sports activities also leads to a decrease in motor competence.

The motivation that is demanded by involvement in active and multisensory approaches based on motor skills and representation (gesture), improves the attention and interest of students towards learning.

Learning through conditions 1 (use of gestures) and 2 (combination of gestures and motor activity) produces better results in the learning of English vocabulary at an early age. As learning was greater in condition 2 than in condition 1, this shows that such approaches are more effective than traditional learning approaches. 

Motivation of students receiving these approaches was high, though this cannot be demonstrated objectively through statistical data. The results revealed that the children liked the condition in which the teaching method was used that combined words, physical activity, and gestures. They affirmed that they would choose this approach as their preferred way of being taught in the future.

The results are in line with previous findings regarding the interaction between physical activity and learning. As mentioned in the introduction, physical activity during childhood not only leads to better academic achievement but also establishes the foundations for engagement in further healthy habits. This is crucial in the present day where a sedentary lifestyle has become the norm for many children.

The present study also supports theconclusions of the study conducted by Toumpaniari et al. [22]. This affirmsthat physically active school programs such as that delivered in Energizers, can be a useful resource for teachers even when teaching new material at a very young age. The incorporation of physical activities in the classroom holdsgreat promise as a way to improve physical health whilst also benefittingchildren’s learning.

We can conclude that motor and expressive activities are an effective motivational resource in children aged 4 to 7 years old from disadvantaged educational settings. When used correctly within the school environment, they canpromote increases in the amount of physical activity engaged in, whilst also improving academic performance and producing a more effective learning of the vocabulary of a second language (English).

## Figures and Tables

**Figure 1 behavsci-09-00084-f001:**
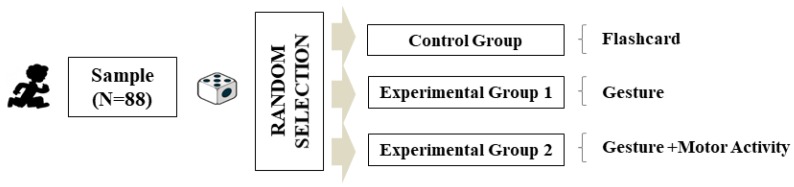
Sample and assignment of study groups.

**Figure 2 behavsci-09-00084-f002:**
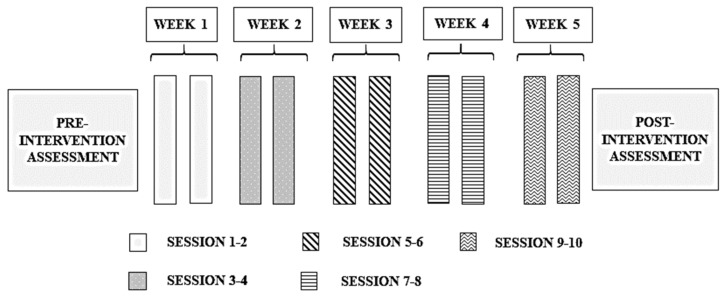
Phases of the intervention program.

**Table 1 behavsci-09-00084-t001:** Data of the Sample.

Gender	
**Man 45.5% (N = 40)** **Woman 54.5 % (N = 48)**	
**Age**	
N = (88) M = 5.45 D.T. = 0.693	
**Family Structure**	
Nuclear Family Single parent (father) Single Parent (mother) Extended family (grandparents)	47.7% (N = 42) 4.5% (N = 4) 34.1% (N = 30) 13.6% (N = 12)
**GameZone**	
Street	34.1% (N = 30)
House	65.9% (N = 58)

**Table 2 behavsci-09-00084-t002:** Vocabulary known by children prior to the intervention commencing (pretest).

Teaching Methodology				
Words	Control Condition	Gestures	Gestures+MotorActiv.	Total
**(0 to 6)**	31.8% (N = 28)	25.0% (N = 22)	43.2% (N = 38)	100.0%

**Table 3 behavsci-09-00084-t003:** Intervention group and number of words assimilated (post-test).

Learned Words					
		(0 to 6)	(7 to 13)	(14 to 22)	*Total*
**Control Condition**	% Students % Words	21.4% (N = 6) 60.0%	57.1% (N = 16) 47.1%	21.4% (N = 6) 13.6%	100% (N = 28) 31.8%
**Gestures**	% Students % Words	18.2% (N = 4) 40.0%	27.3% (N = 6) 17.6%	54.5% (N = 12) 27.3%	100% N = 22) 25.0%
**Gestures+** **Motor Activity**	% Students % Words	0.0% (N = 0) 0.0%	31.6% (N = 12) 35.3%	68.4% (N = 26) 59.1%	100% (N = 38) 43.2%

**Table 4 behavsci-09-00084-t004:** The approach used with degree of satisfaction reported by the sample.

**Question 1**			
	**Yes**	**I don’t know**	**No**
**Control Condition**	92.9%	7.1%	0%
**Gestures**	90.9%	9.1%	0%
**Gestures+ Motor Activity**	100.0%	0.0%	0%
**Question 2**			
**Control Condition**	92.9%	7.1%	0%
**Gestures**	100.0%	0.0%	0%
**Gestures+ Motor Activity**	100.0%	0.0%	0%

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
