# Peer review of "Effectiveness of a Motor Intervention Program on Motivation and Learning of English Vocabulary in Preschoolers: A Pilot Study"

_behavsci, 2019, doi:10.3390/bs9080084_

Round 1
Reviewer 1 Report
General, the biggest concern for me the methodology section and limitations that are associated with that. I have not answered questions on novelty or significance of content as my expertise in this population group is not what is needs to be to make a judgement on that. I wish you the best with your work.
Introduction
Geographical and international consideration for context for past research lines 38-44
Line 66-92 seems to represent an overview but is descriptive and you bring together past studies and clearly identify the gaps within the literature – clearly state what is known, not known and further needs consideration across the past set of studies – point the reader towards why you are using the outcome assessments that you are and why you are selecting the groups that you are. Think about confounders and if this has an implication for your analysis choice etc the discussion gives some more focused consideration to your selected population group and past work
Methods
You need to identify a trial registration number and protocol and if any changes to the protocol were made
Trail design comes first:
The abstract and methods says longitudinal ? need to mention that it was parallel groups and the allocation ratio –
also if it was not randomized you need to identified this as well as allocation concealment and blinding or identify it as limitations.
You need to mention sample size calculation
You need a specific section on eligibility criteria
There is a danger with you lack of outcome measure that you miss important confounders e.g., identified in your dicussion
Ethical approval number needed
Is the setting clear ?
Consider the need to use the Tider guidelines for intervention documentation
Can you identify an ethical approval number and committee
Re-consider analysis for three groups – I don’t think t test would be the choice and if it was you need to adjust you p value
Results
Need to identify where the difference is and provide detail of the difference including effect size, confidence interval of the difference etc
Discussion
Start off with a greater consideration to the findings
Seems to be some more and focused consideration here to the group of interest
Author Response
Thank you very much for your time and dedication with us
Dear editor, firstly we would like to thank you for the feedback of your review. We also apologize for the mistakes in the methodical section.
We have tried to answer and to improve all your suggestions. Please, check all the improvements (in red) that we have incorporated, pointing the lines where they can be found.
Hopefully, we have understood all the suggestions properly and we have improved the paper for further publication.
Thank you very much for your time and support
Introduction
Geographical and international consideration for context for past research lines 38-44
New international studies have been included that affirm the absence of physical activity classes and programs at preschool age. The following bibliographical citations are included (Lines 49-50).
Line 66-92 seems to represent an overview but is descriptive and you bring together past studies and clearly identify the gaps within the literature – clearly state what is known, not known and further needs consideration across the past set of studies – point the reader towards why you are using the outcome assessments that you are and why you are selecting the groups that you are. Think about confounders and if this has an implication for your analysis choice etc the discussion gives some more focused consideration to your selected population group and past work
The background of studies thathaveimplementedmethodologiesthat use motor skills at earlyages, and therelevantconclusionsoneachone, both for and against, are collected (Lines 77-129).
Methods
An in-depth review of the methodology has been carried out. Solving the errors and adding all the indicated parts:
You need to identify a trial registration number and protocol and if any changes to the protocol were made
The process of coding the sample is described to preserve the confidentiality of the data and the simple (194-195).
Traildesign comes first:
The abstract and methods says longitudinal? need to mention that it was parallel groups and the allocation ratio – also if it was not randomized you need to identified this as well as allocation concealment and blinding or identify it as limitations.
The study desing was corrected in the lines 12 and 144-146.
You need to mention sample size calculation and specific section on eligibility criteria
The sample has been selected for convenience, corresponding to the total number (100%) of the population of centers.
They have been added in the lines 155-162
There is a danger with you lack of outcome measure that you miss important confounders e.g., identified in your dicussion
Lines 291-307 aim to improve the contextualization of the study based on previous studies
Consider the need to use the Tider guidelines for intervention documentation
I am very grateful for the suggestion, I did not know the Tider guidelines and I find it very useful to improve future works. Thank you very much for the indication
Can you identify an ethical approval number and committee// Ethical approval number needed. You canfind it in the lines 180-186
Re-consider analysis for three groups – I don’t think t test would be the choice and if it was you need to adjust you p value
The study desing was corrected in the lines 12 and 144-146.
Results
Need to identify where the difference is and provide detail of the difference including effect size, confidence interval of the difference etc
The type of data analysis was wrong. We feel the mistake made. We have modified the information, adding the correct analyzes in lines 250-254.
Discussion
Start off with a greater consideration to the findings .Seems to be some more and focused consideration here to the group of interest
The limitations of the study are in lines 386-398.
This information has been added in the paragraphs 256-262 y 283-284.
Attempts have also been made to improve the conclusions and the limitations of the study have been added.
Conclusions
Lines 418-431 complete some aspects of the conclusions obtained
Bibliographic references
New references are included (28,29,32,33,67)

Reviewer 2 Report
The paper presents a structure and correct methodology.
The methodological designperformed shows evidence of the quality of work.
Furthermore, the extensive literature review, which cites more than 70 references to major international bibliographic sources is one of the strengths of this work.
However, a final review of the literature may be necessary to include some important papers published about this subject that are not cited in this paper.
Author Response
Introduction
Geographical and international consideration for context for past research lines 38-44
New international studies have been included that affirm the absence of physical activity classes and programs at preschool age. The following bibliographical citations are included (Lines 49-50).
Line 66-92 seems to represent an overview but is descriptive and you bring together past studies and clearly identify the gaps within the literature – clearly state what is known, not known and further needs consideration across the past set of studies – point the reader towards why you are using the outcome assessments that you are and why you are selecting the groups that you are. Think about confounders and if this has an implication for your analysis choice etc the discussion gives some more focused consideration to your selected population group and past work
The background of studies thathaveimplementedmethodologiesthat use motor skills at earlyages, and therelevantconclusionsoneachone, both for and against, are collected (Lines 77-129).
Methods
An in-depth review of the methodology has been carried out. Solving the errors and adding all the indicated parts:
You need to identify a trial registration number and protocol and if any changes to the protocol were made
The process of coding the sample is described to preserve the confidentiality of the data and the simple (194-195).
Traildesign comes first:
The abstract and methods says longitudinal? need to mention that it was parallel groups and the allocation ratio – also if it was not randomized you need to identified this as well as allocation concealment and blinding or identify it as limitations.
The study desing was corrected in the lines 12 and 144-146.
You need to mention sample size calculation and specific section on eligibility criteria
The sample has been selected for convenience, corresponding to the total number (100%) of the population of centers.
They have been added in the lines 155-162
There is a danger with you lack of outcome measure that you miss important confounders e.g., identified in your dicussion
Lines 291-307 aim to improve the contextualization of the study based on previous studies
Consider the need to use the Tider guidelines for intervention documentation
I am very grateful for the suggestion, I did not know the Tider guidelines and I find it very useful to improve future works. Thank you very much for the indication
Can you identify an ethical approval number and committee// Ethical approval number needed. You canfind it in the lines 180-186
Re-consider analysis for three groups – I don’t think t test would be the choice and if it was you need to adjust you p value
The study desing was corrected in the lines 12 and 144-146.
Results
Need to identify where the difference is and provide detail of the difference including effect size, confidence interval of the difference etc
The type of data analysis was wrong. We feel the mistake made. We have modified the information, adding the correct analyzes in lines 250-254.
Discussion
Start off with a greater consideration to the findings .Seems to be some more and focused consideration here to the group of interest
The limitations of the study are in lines 386-398.
This information has been added in the paragraphs 256-262 y 283-284.
Attempts have also been made to improve the conclusions and the limitations of the study have been added.
Conclusions
Lines 418-431 complete some aspects of the conclusions obtained
Bibliographic references
New references are included (28,29,32,33,67)
Round 2
Reviewer 1 Report
Just small comments – mainly early on just to bring it together further.
Abstract
Consider the lines around design you say on line 12-13 “A quasi-experimental study was carried out, with an experimental(pretest-posttest) design in a sample (n=88)” confusing because you say quasi then experimental – can you remove the word experimental and replace with what you say in your methods section (which is clearer) “A quasi-experimental design study was carried out using a pretest-posttest with a non-equivalent control group”
Line 80 needs ] in the reference 51.
After line 80 you can tell the reader an overview of what is coming perhaps? Like 7 methods for learning have been identified: (a) good start method, use your text, (b) methods based rhythmic…
Use of bullet points between line 77-103 needs consideration are they needed should there be seprate paragraphs if so?
Line 96-115 are in bold which can be removed
Line 100 full stop needed. Line 104 you state: any content (directions, animals, health, languages, etc.) – can you be more accurate e.g., content based on 7 domains (directions, animals, health) – just tell the reader – think about informative nature of statements
Separate out the detail on the methods from the final. Summary paragraph is needed from around 110 so bring together
Not sure you need information from line 121 is it methods?
Line 126 to line 137 contains results move to that section here you can name design and eligbility criteria
Author Response
Dear Reviewer:
Thank you very much for your time and dedication with us
Dear editor, firstly we would like to thank you for the feedback of your review.
We have tried to answer and to improve all your suggestions. Please, check all the improvements (in red) that we have incorporated, pointing the lines where they can be found.
We have reviewed the changes and suggestions of the second round taking into account the version that is registered on the platform. This version is the last one we have sent and does not match the lines with the first version, on which you have given us the indications.
Hopefully, we have understood all the suggestions properly and we have improved the paper for further publication.
Thank you very much for your time and support
Abstract
Consider the lines around design you say on line 12-13 “A quasi-experimental study was carried out, with an experimental (pretest-posttest) design in a sample (n=88)” confusing because you say quasi then experimental – can you remove the word experimental and replace with what you say in your methods section (which is clearer) “A quasi-experimental design study was carried out using a pretest-posttest with a non-equivalent control group”
Thanks for the indication, it was a mistake not to change it (Line 13)
Line 80 needs ] in the reference 51.
Thanks for the indication, it was a mistake (line 51)
After line 80 you can tell the reader an overview of what is coming perhaps? Like 7 methods for learning have been identified: (a) good start method, use your text, (b) methods based rhythmic…
Use of bullet points between line 77-103 needs consideration are they needed should there be seprate paragraphs if so?
Like 5 methods for learning have been identified : a)…..b)….c)….d)…e)… (Line 81-110)
Line 96-115 are in bold which can be removed
I do not understand well. For the improvement of translation and text, change control has been used. The changes suggested in the text should be accepted for improvement.
Line 100 full stop needed. Done, line 106
Line 104 you state: any content (directions, animals, health, languages, etc.) – can you be more accurate e.g., content based on 7 domains (directions, animals, health) – just tell the reader – think about informative nature of statements
It refers to the fact that any curricular content can be worked on. The parenthesis has been changed to curricular content. (Line 110)
Separate out the detail on the methods from the final. Summary paragraph is needed from around 110 so bring together . Not sure you need information from line 121 is it methods?
After reading we have removed line (130), as suggested.
Line 126 to line 137 contains results move to that section here you can name design and eligbility criteria
Data regarding the results of the sample have been deleted (lines 133-135).